



# Analyzing the Atmospheric Boundary Layer by high-order moments obtained from multiwavelength lidar data: impact of wavelength choice

Gregori de Arruda Moreira[1,2,3,4], Fábio Juliano da Silva Lopes[4], Juan Luis Guererro-Rascado[1],
Jonatan João da Silva[4,5], Antonio Arleques Gomes[4], Eduardo Landulfo[4], and Lucas Alados-Arboledas[1]

[1]Andalusian Institute for Earth System Research, Granada, Spain
[2]Dpt. Applied Physics, University of Granada, Granada, Spain
[3]Astronomy, Geophysics and Atmospheric Science Institute, University of São Paulo, São Paulo, Brazil
[4]Nuclear and Energy Research Institute, São Paulo, Brazil
[5]Federal University of Western Bahia, Bahia, Brazil

**Correspondence:** Gregori de Arruda Moreira (gregori.moreira@usp.br)

**Abstract.** The lowest region of the troposphere is a turbulent layer denominated Atmospheric Boundary Layer (ABL) characterized by high daily variability due to the influence of surface forcings. This is the reason why detecting systems with high spatial and temporal resolution, like lidars, have been widely applied for researching this region. In this paper, we present a comparative analysis on the use of lidar backscattered signals at three wavelengths (355, 532 and 1064 nm) to study the ABL

5   investigating the high-order moments, which give us information about the ABL height (derived by the variance method), aerosol layers movements (skewness) and mixing conditions (kurtosis) at several heights. Previous studies have shown that 1064-nm wavelength, due to the predominance of particle signature in the total backscattered atmospheric signal, provides an appropriate description of the turbulence field and thus, in this study, it was considered as a reference. We analyze two case studies, which show us that the backscattered signal at 355 nm, even after applying some corrections, has a limited applica-

10  bility for turbulence studies using the proposed methodology due to the strong contribution of the molecular signature to the total backscatter signal. This increases the noise associated to the profiles and, consequently, generates misinformation. On the other hand, the information on the turbulence field derived from the backscattered signal at 532 nm is similar to that obtained at 1064 nm due to the appropriate attenuation of the noise, generated by molecular component of backscattered signal, by the application of the corrections proposed.

## 1   Introduction

The Atmospheric Boundary Layer is the part of the troposphere that is directly or indirectly influenced by the Earth's surface (land and sea), and responds to gases and aerosol particles emitted at the Earth's surface and to surface forcing at time scales





less than a day. Forcing mechanisms include heat transfer, fluxes of momentum, frictional drag and terrain-induced flow modification. The height of this layer (ABLH) varies from hundreds of meters until some kilometers due to the intensification or reduction of convective or mechanical processes with additional contribution from orographic effects. The *ABL* presents a daily pattern controlled by the energy balance at the Earth's surface. Thus, after sunrise the positive net radiative flux ($R_n$)

induces the raise of air surface temperature that initiates the convective process, which is responsible for the growth of the so-called Mixing Layer (ML) or Convective Boundary Layer (CBL). This layer grows along the day extending the region affected by the convective process until around midday, when it reaches their maximum development. Slightly before sunset, the decrease of the incoming solar irradiance incoming at the surface goes towards negative values and thus, corresponding to radiative cooling of the Earth's surface. This cooling affects the closest air layer, diminishing the convective process. In this

way, the CBL disappears and two new layers characterize the ABL, a stable and stratified layer denominated Stable Boundary Layer (SBL) at the bottom and the Residual Layer (RL) over the last one with characteristics of the previous day's ML (Stull, 1988).

The turbulent features of the ABL are relevant in air quality and weather forecasting and thus are worthy of study. As a rule, the turbulent processes are treated as nondeterministic and, therefore, the turbulence is characterized by its statistical

properties. Thus, high order statistical moments are used to generate information about the turbulent fluctuation field, besides a description about mixing processes in the ABL (Pal et al., 2010).

ABL turbulence has been commonly studied by means of anemometer towers (e.g., Kaimal and Gaynor, 1983) and aircrafts (e.g., Lenschow et al., 1980; Williams and Hacker, 1992; Lenschow et al., 1994; Stull et al., 1997; Andrews et al., 2004; Vogelmann et al., 2012). Nevertheless, the first ones have a use restricted to regions near the surface, due to their limited vertical

range. Aircrafts offer an alternative approach that allows extending the analyses to higher atmospheric layers, but conversely, they have a reduced time window, thus limiting the period of analysis. Due to the large variability of the ABL characteristics along the day, the use of systems endowed with high spatial and temporal resolution allow studies with a higher degree of details. Consequently, remote sensing systems (mainly lidars) become an important tool in turbulence studies (Lagouarde et al., 2013, 2015). In addition, the different lidar techniques offer the possibility of analyses with several variables, such as

vertical wind velocity by Doppler lidar (Lenschow et al., 2000; Lothon et al., 2006; O'Connor et al., 2010), water vapor mixing profile by Raman lidar or Differential Absorption lidar (DIAL) (Wulfmeyer, 1999; Kiemle et al., 2007; Wulfmeyer et al., 2010; Turner et al., 2014; Muppa et al., 2016), temperature by rotational Raman lidar (Behrendt et al., 2015) and aerosol number density by elastic lidar or High Spectral Resolution lidar (HSRL) (Pal et al., 2010; McNicholas and Turner, 2014). Therefore, a wider range of results can be obtained, especially when different types of systems are synergistically used, as shown by

Engelmann et al. (2008) who combine elastic and Doppler lidar data for deriving the vertical aerosol flux.

Pal et al. (2010) have shown that it is feasible the use of elastic lidar measuring at a high acquisition rate for characterizing the atmospheric turbulence. In particular, they have shown that the fluctuation of the Range Corrected Signal (RCS) at 1064 nm is a proxy for the fluctuation of the particle concentration, due to predominance of particle signature ($\beta_{par}$) in the total backscattered signal at this wavelength, and, thus, it can be used for observing the turbulent aerosol movements in the CBL.

However, if other wavelengths are used in this kind of analysis, the effects of molecular backscatter coefficient ($\beta_{mol}$) and





atmospheric extinction ($\alpha$) must be considered. In this work, we perform a comparative analysis regarding the use of three different wavelengths, namely 355, 532 and 1064 nm (the last one adopted as reference), to obtain the high-order moments, i.e. variance ($\sigma^2$), skewness ($S$) and kurtosis ($K$), and also the integral time-scale ($\tau$). Moreover, it was analyzed the interference of noise $\varepsilon$ and $\beta_{mol}$ over the high-order moments and $\tau$ obtained from each one of the considered wavelengths, in order

to quantify how such factors can influence the correct interpretation of the statistical variables. The goal of this study is to show the viability of the proposed methodology for studying the turbulence by computing the high-order moments of the backscattered signal at different wavelengths. We pay special attention to the advantages and limitations of each wavelength analyzed considering the importance of the proposed correction schemes. This paper is organized as follows. The measurement site and the experimental set up are introduced in Section 2. The methodology is described in Section 3. The comparisons and

case studies are analyzed in Section 4. Conclusions are given in Section 5.

## 2   Experimental site and instrumentation

This study was performed at LEAL (Laser Environmental Applications Laboratory) from July 2017 to July 2018. LEAL is part of the Latin America Lidar Network - (Guerrero-Rascado et al., 2016; Antuña Marrero et al., 2017) since 2001. This lidar facility is located at installed at the Nuclear and Energy Research Institute in São Paulo-Brazil (23°33'S, 46°38'W,

760 m a.s.l.), and it is the largest metropolitan area in South America, with a population of approximately 12 million of inhabitants, and endowed a subtropical climate where winter is mild (15°C) and dry, while summer is wet and has moderately high temperatures (23°C) (IBGE, 2017). The SPU Lidar station has a coaxial ground-based multiwavelength Raman lidar system operated at LEAL. The system operates with a pulsed Nd:YAG laser, emitting radiation at 355, 532 and 1064 nm emitting radiation at 355, 532 and 1064 nm, with a laser repetitive rate of 10 Hz and beam pointing perpendicular to surface.

The pulse energy and stability of each wavelength are 225 mJ and 2 (355 nm), 400 mJ and 4 (532 nm), 850 mJ and 6 (1064 nm), respectively. The SPU Lidar station detects three elastic channels at 355, 532 and 1064 nm and three Raman-shifted channels at 387 (from $N_2$), 408 (from $H_2O$), both when excited with the wavelength of 355 nm, and 530 nm (from $N_2$ when excited with the wavelength of 532 nm). The SPU lidar station reaches full overlap at around 300 m a.g.l. (Lopes et al., 2018). This system was operated with a temporal and spatial resolutions of 2 s and 7.5 m, respectively.

## 3   Methodology

The turbulence study is based on the observation of the fluctuation $q'(t)$ of a determined variable (q) in the time $t$. The values are obtained as follows: firstly $q(t)$ are averaged in packages that cover a certain time interval, from which the mean value ($\bar{q}$) is extracted. Then, such value is subtracted from each $q(t)$ value, providing the fluctuation $q'(t)$ as demonstrated in the equation below by Reynold's decomposition (de Arruda Moreira et al., 2019):

$$q'(t) = q(t) - \bar{q}(t) \tag{1}$$

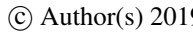



In the analysis performed with elastic lidar systems the variable of interest is the aerosol number density ($N$) and thus its fluctuation ($N'$) can be obtained by equation 1. However, elastic lidar systems do not directly provide $N$. Then, it is necessary to assume some considerations to estimate this variable, which were described by (Pal et al., 2010). Ignoring the hygroscopic growth and assuming similar types of aerosol throughout the atmospheric column, the following equation can be used:

$$\beta_{aer}(z,t) \approx N(z,t)Y(z) \tag{2}$$

$$\beta'_{aer}(z,t) = N'(z,t) \tag{3}$$

where $\beta_{aer}$ and $\beta'_{aer}$ represent the particle backscatter coefficient and its fluctuation, respectively. The variable $z$ is the height above the ground, $t$ is the time and $Y$ is a variable that does not depend on time.

The lidar equation is defined as follows Weitkamp (2005):

$$P(z,t) = P_0 \frac{c\tau}{2} A\eta O(\lambda,z)\frac{\beta(\lambda,z)}{z^2} \, exp\left[-2\int_o^z \alpha(\lambda,\,z')\mathrm{d}z'\right] \tag{4}$$

where $P(\lambda,\,z)$ is the power signal [W] detected at a distance $z$ [m] and time $t$ [s], $z$ is the distance [m] of the atmospheric volume investigated, $P_0$ is the power emitted by the laser source [W], $c$ is the speed of light [m/s], $\tau$ the laser pulse duration [ns], and $A$ is the effective area of the telescope receptor [$m^2$], $\eta$ is a variable related to the efficiency of the lidar system

and $O(\lambda,z)$ is the laser-beam receiver-field-of-view overlap function. The most important quantities are $\beta(\lambda,z)$, which is the total backscatter coefficient, due to atmospheric molecules, $\beta_{mol}(\lambda,z)$, and aerosol $\beta_{aer}(\lambda,z)$, in other words, $\beta(\lambda,z) = \beta_{mol}(\lambda,z) + \beta_{aer}(\lambda,z)$ [$(m.sr)^{-1}$] at distance $z$, and $\alpha(\lambda,z)$ is the total extinction coefficient, due to atmospheric molecules, $\alpha_{mol}(\lambda,z)$, and aerosols $\alpha_{aer}(\lambda,z)$, in other words, $\alpha(\lambda,z) = \alpha_{mol}(\lambda,z) + \alpha_{aer}(\lambda,z)$ [$(m)^{-1}$] at distance $z$. If the wavelength 1064 nm is used, we can neglect the influence of the extinction coefficient $\alpha(\lambda,z)$ provided by aerosol, the Rayleigh scattering

generated by atmospheric molecules) and the $\beta_{mol}(\lambda,z)$ (Pal et al., 2010). Therefore, the equation 4, for the wavelength of 1064 nm, can be rewritten as follows:

$$RCS_{1064}(z,t) = P_{1064}(z,t) \cdot z^2 \approx G \cdot \beta_{1064}(z,t) \approx G \cdot \beta_{aer}(z,t) \tag{5}$$

where $RCS_{1064}$ is the Range Corrected Signal, $G$ is a constant and the subscribed indexes represent the wavelength and the particles. Then, for 1064 nm:

$$RCS'_{1064}(z,t) \approx \beta'_{1064}(z,t) = \beta'_{aer}(z,t) = N'(z,t) \tag{6}$$





Our purpose is to evaluate the use of other wavelengths when the effects of molecular backscatter coefficient ($\beta_{mol}$). The interest is based on the best performance of the technology for detecting wavelengths in the VIS and UV and on the extended use of these wavelengths in the lidar networks: The Latin America Lidar Network - LALINET (Guerrero-Rascado et al., 2016; Antuña Marrero et al., 2017), European Aerosol Research Lidar Network – EARLINET (Pappalardo et al., 2014) and The
NASA Micropulse Lidar Network – MPLNet (Welton et al., 2001).

### 3.1  High-order moments

The high-order moments used in this study are obtained from $RCS'(z,t)$, generated by equation 1, where $\overline{RCS}(z)$ represents the 1-hour average package of $RCS(z,t)$ data. From this, the high order moments, variance ($\sigma^2$), skewness ($S$) and kurtosis ($K$) are obtained as demonstrated in the first column of Table A1 , as well as, their corrections and errors in the second and
third columns of the same table, respectively. In table A2 are presented the physical meaning of each high-order moment in the context of the proposed analysis

The integral time scale ($\tau$) is an important prerequisite in turbulence studies. It guarantees that the most part of the horizontal variability of the turbulent eddies is detected with good resolution, enabling the solution of inertial subrange and dissipation range in the spectrum and autocorrelation function, respectively (Pal et al., 2010). $\tau$ must be larger than the temporal resolution
of the analyzed time series (SPU Lidar station time acquisition is 2s). In the same way of high-order moments, such variable is obtained from $RCS'(z,t)$ as shown in the first column of Table A1.

### 3.2  Error analysis

The high-order moments and $\tau$ generated from $RCS'(z,t)$ can also be obtained from the following autocovariance function $M_{ij}$, which has its order represented by the sum of the subscript i and j (Pal et al., 2010), according to the following equation:

$$
\quad M_{ij} = \int_{0}^{t_f} \left[ RCS'(z,t) \right]^i \left[ RCS'(z,t+t_f) \right]^j dt \tag{7}
$$

However, it is important to consider the influence of instrument noise $\varepsilon(z,t)$ in the $RCS'(z,t)$ profile. Therefore, $M_{ij}$ can be rewritten as follows:

$$
M_{ij} = \int_{0}^{t_f} \left[ RCS'(z,t) + \varepsilon(z,t) \right]^i \left[ RCS'(z,t+t_f) + \varepsilon(z,t+t_f) \right]^j dt \tag{8}
$$

Although atmospheric fluctuations are correlated in time, $\varepsilon(z,t)$ is random and uncorrelated with the atmospheric signal,
therefore $\varepsilon(z,t)$ is only associated with lag 0. Consequently, it is possible to obtain the corrected autocovariance function,



$M_{11}(\to 0)$, removing the error $\Delta M_{11}(0)$ of the uncorrected autocovariance function $M_{11}(0)$, as demonstrated in the equation below:

$$M_{11}(\to 0) = M_{11}(0) - \Delta M_{11}(0) \tag{9}$$

Based on this concept, Lenschow et al. (2000) proposed two methods to correct for the noise influence:

- First lag correction: the lag 0 ($\Delta M_{11}(0)$) is directly subtracted from the uncorrected autocovariance function $M_{11}(0)$, generating $M_{11}(\to 0)$.

- -2/3 law correction: A new lag 0 value is obtained by the extrapolation of $M_{11}(0)$ to the firsts non-zero lags back to lag zero, using the inertial subrange hypothesis (Monin and Yaglom, 2013):

$$M_{11}(\to 0) = \overline{RCS'(z,t)} + Ct^{2/3} \tag{10}$$

where $C$ represents a parameter of turbulent eddy dissipation rate. In this study, we also used the first five points after lag 0 to perform this correction. In Table A1 the second and third columns present the corrections and errors, respectively, of high-order moments and $\tau$.

Figure A1 shows how the procedures described in section 3.1 and 3.2 are used. Firstly, the lidar data are acquired with time resolution of two seconds. Then, these data are averaged in packages of one-hour generating $\overline{RCS(z)}$, from which is possible to obtain $RCS'(z,t)$ as illustrated in equation 1. Then, the two corrections shown in section 3.2 are separately applied. Finally, the high-order moments and the $\tau$, corrected and without correction, are estimated.

## 4    Results

In this section we present two case studies, applying the methodology described in section 3, in order to perform a comparative analysis about the influence of $\beta_{mol}$, and $\varepsilon$ in the high-order moments and $\tau$ obtained from different wavelengths (355, 532 and 1064 nm).

### 4.1    Case Study I: $26^{th}$ July 2017

In this case study we gathered measurements from 13:00 to 19:00 UTC. Figure A2 shows the time-height plot of $RCS_{532}$ during this period. This case is composed by two distinct periods, in the first two hours there is a RL with an underlying shallow CBL. Nevertheless, in the last part of the second hour the CBL quickly grows and it mixes with RL forming a fully-developed ABL, with its top situated between 1500 and 1600 m from 15:00 to 19:00 UTC. The black dotted box, between 17:00 and 18:00 UTC represents the period selected to perform the statistical analysis.





In order to check the hypothesis proposed by (Pal et al., 2010), which assumes that there is not particle hygroscopic growth and that the same type of aerosol is present in the entire atmospheric column, were analyzed the relative humidity and mixing ratio profile retrieved from radio-sounding measurements (http://weather.uwyo.edu/upperair/sounding.html), launched at the Campo de Marte Airport (São Paulo, Brazil), which is about 10 km away from the SPU lidar system. Figure A3-A and

A3-B shows the relative humidity and mixing ratio profiles, respectively, measured on $26^{th}$ July 2017 at 12 UTC. Both, relative humidity and mixing ratio can be considered constant below 1500 m, with mean values of $67 \pm 8\%$ and $7.6 \pm 0.9\,g/kg$, respectively. Since there are no large variation of water vapor mixing ratio and relative humidity values in this region, we assume that this case is not affected by particle hygroscopic growth. In addition, the AERONET Sunphotometer (Holben et al., 1998a) data from the São Paulo station were retrieved in order to check the aerosol type, as can be seen in the figure A3-C.

According to Eck et al. (1999), the Ångström Exponent (AE) can be a useful tool to distinguish different types of atmospheric aerosols. Figure A3-C shows the aerosol AE time series for the case study of $26^{th}$ July 2017. The AE was calculated at the spectral range 340-440 nm and 440-675 nm using AERONET (Holben et al., 1998b) products from Level 1.5 version 3 data. For this measurement period the percentage variation of AE was no more than 3% in both cases. Therefore, there are no considerable changes during the whole measurement period, which is a strong indication that there is no aerosol type change

throughout the day.

In figure A4 is presented the Signal-to-Noise-Ratio (SNR) profile of the raw lidar signal, as calculated by Heese (2010), of the three wavelengths (1064 nm (red line), 532 nm (green line), and 355 nm (violet line)) during the analyzed period. All wavelengths have values of SNR higher than 1 (the threshold for good quality) below the ABLH (dotted blue line) with predominance of values lower than 1 in the FT, what was expected due to the strong reduction of aerosol concentration in such

region. Although the three wavelengths have similar SNR profiles, close to ABLH the difference among then become more evident, principally the fast decreasing of the 355 nm and the high values of 532 nm.

Figure A5 shows the autocovariance function (ACF), obtained between 17:00 and 18:00 UTC for the wavelengths 355 ($ACF_{355}$), 532 ($ACF_{532}$) and 1064 nm ($ACF_{1064}$) at 1000 m agl and 1700 m agl. Thus, from the Figure A2 it is possible to observe that the first height is situated below the top of CBL and the last one at FT. As expected $\varepsilon$ increases with height for

all the wavelengths due to reduction of aerosol load with height. $ACF_{355}$ has the lowest intensity (around 90% smaller those of $ACF_{532}$ and $ACF_{1064}$ ) and it is clearly much more affected by the magnitude of $\varepsilon$ that represents approximately 25% of $ACF_{355}$, while for $ACF_{532}$ and $ACF_{1064}$ the noise represents around 10% of the respective autocovariance.

Figure A6 presents all statistic variables, their respective corrections and errors (shadows), generated from the methodology described in section 3, for data acquired between 17:00 and 18:00 UTC.

The variance profiles, $\sigma^2_{RCS}(z)$, with and without corrections for all wavelengths are represented in Figure A6, from 1 to 9. The low and almost constant values of uncorrected $\sigma^2_{RCS_{1064}}(z)$ from the bottom until around 1000 m of altitude demonstrates an almost constant distribution of aerosol particles in this region, as can be seen in Figure A6.1. Above 1000 m of altitude, the value of uncorrected $\sigma^2_{RCS_{1064}}(z)$ increases, reaching its maximum peak at around 1600 m. This peak represents the Entrainment Zone, the region where a mixing occurs between air parcels coming from the CBL and FT. According to Menut

et al. (1999), there is an intense variation of aerosol concentration during this process, generating a maximum in the uncorrected



$\sigma^2_{RCS_{1064}}(z)$, which represents the *ABLH*. Above the *ABHL*, the aerosol concentration is considerably lower than in *CBL* and, thus, the uncorrected $\sigma^2_{RCS_{1064}}(z)$ is reduced to practically zero. This methodology to estimate the ABLH is named Variance Method or Centroid Method and it was described by Hooper and Eloranta (1986) and Menut et al. (1999), respectively. The main limitations of this method are its applicability only for CBL, and the ambiguous results in complex cases, like as the

presence of several aerosol layers Emeis (2011). In such situations more sophisticated methods like as Wavelet Pal et al. (2010), PathfinderTURB Poltera et al. (2017) and POLARIS Bravo-Aranda et al. (2017) are recommended.

The uncorrected $\sigma^2_{RCS_{532}}(z)$, presented in Figure A6.4 is rather similar to uncorrected $\sigma^2_{RCS_{1064}}(z)$, including the position of maximum peak. Nevertheless, although uncorrected $\sigma^2_{RCS_{355}}(z)$, presented in Figure A6.7, also has the maximum peak situated at around 1600 m of altitude, the profile is nosier than the profiles obtained from the other wavelengths and, therefore,

it is not possible to identify the regions with uniform aerosol distribution as evidenced in uncorrected $\sigma^2_{RCS_{1064}}(z)$.

The correction 2/3, shown in Figures A6.2, A6.5 and A6.8, does not cause significant changes in the uncorrected profiles. On the other hand, the first lag correction changes significantly the profiles, thus $\sigma^2_{RCS_{532}}(z)$ becomes very similar to $\sigma^2_{RCS_{1064}}(z)$, while $\sigma^2_{RCS_{355}}(z)$ continues with some differences, mainly in the region below the *ABLH*, as can be seen in Figures A6.3, A6.6 and A6.9.

The integral time scale profiles $\tau_{RCS'}(z)$, with and without corrections $\tau^{corr}_{RCS'}(z)$ and $\tau^{unc}_{RCS'}(z)$, respectively, calculated for the three wavelengths are presented in the Figure A6, from 10 to 18. The $\tau^{unc}_{RCS'}(z)$ presents values larger than SPU Lidar station time acquisition showed as black dotted line, in the region below *ABLH* at all wavelengths, as can be seen in Figures A6.10, A6.13 and A6.16. The largest values of $\tau^{unc}_{RCS'}(z)$ correspond to 1064 nm, while the lowest values are computed for 355, which is practically half of those obtained with the reference wavelength, 1064 nm. The low value for the $\tau^{unc}_{RCS'}(z)$ at 355

20  nm can be associated to the influence of the noise in the signal retrieved at this wavelength. The application of the correction 2/3 does not cause significant changes in the profiles, while the first lag correction changes significantly the profiles mainly in the region below the *ABLH*, as can be checked in Figures A6.11, A6.14 and A6.17, and in Figures A6.12, A6.15 and A6.18, respectively.

The skewness profiles $S_{RCS}(z)$ represent the degree of asymmetry in a distribution, where $S_{RCS}(z) = 0$ represents sym-

metric distributions about its mean, while positive and negative values represents cases where the tail of distribution is on the left and right side of the distribution, respectively. The uncorrected skewness profiles $S^{unc}_{RCS}(z)$ and their respective corrections $S^{corr}_{RCS}(z)$, for the three wavelengths are presented in the Figures A6, from 19 to 27. The $S^{unc}_{RCS}(z)$ generated from the wavelengths 1064 and 532 nm, presented in Figures A6.19 and A6.22, respectively, presents similar behavior up to approximately 150 m above the $ABLH_{elastic}$, with positive values in the low part of the profile and one inflection point close to

$ABLH_{elastic}$. Such point characterizes the transition from the region with entrainment of clean *FT* air into the *CBL* (negative values) to a region few meter above the $ABLH_{elastic}$ with presence of aerosol plumes (positive values) due to convective movement. This behavior of skewness profile also was observed by Pal et al. (2010) and McNicholas and Turner (2014) at the region of the $ABLH_{elastic}$. Therefore, the same set of phenomena is evidenced by the dataset at both wavelengths, although there are differences in the absolute values.





The two corrections cause negligible variations in the profiles at 1064 nm, as shown in Figures A6.20 and A6.21. On the other hand, the corrections applied to the $S_{RCS}^{unc}(z)$ at 532 nm produce skewness profiles similar to those at the reference wavelength, as can be checked in Figures A6.23 and A6.24. It is possible to observe a difference between the skewness profiles at 532 nm (positive) and 1064 nm (negative) in the region above the $ABLH_{elastic}$. Such difference is a consequence of the low

values of signal-to-noise-ratio ($SNR$) of the RCS' and consequently $\tau_{RCS}(z)$ observed in this region, preventing the observation of turbulence due to technical limitations of the instruments used. The skewness profiles at 355 nm, $S_{RCS}^{corr}(z)$ and $S_{RCS}^{unc}(z)$, present a rather different behavior and do not follow the same variations observed in the reference wavelength profile, as can be seen in Figures A6.25, A6.26 – 2/3 correction and A6.27 – first lag correction. Consequently, it is not possible to observe the aerosol dynamics using the information gathered at the wavelength 355 nm.

The kurtosis profile $K_{RCS'}$ is the most complex high-order moment presented in this study and, consequently, in such profiles the differences among the three wavelengths are more evident. In the context of our analysis, the values of $K_{RCS'}$ are indicators of the mixing degree at each altitude, as well as, of the intermittence of turbulence caused by large eddies. In reason of some technical limitations of our lidar system, it is possible to resolve eddies only until a predetermined size. Therefore, in regions where turbulence is performed in too small scales, our system cannot solve these eddies. Values lower than 3 represents

a well-mixed region, indicating a flatter distribution in comparison with a normal distribution, thus the turbulence caused by large eddies can be characterized as frequent. In contrast, values higher than 3 indicates a peaked distribution in comparison with a Gaussian distribution, in other words, there is an unusual variation in the $RCS'(z,t)$, which represents a low degree of mixing, and the presence of an infrequent large eddies turbulence (Pal et al., 2010).

The $K_{RCS'}^{unc}$ at 532 and 1064 nm have some differences in the region below 1300 m of altitude, where the profile at 1064 nm

only shows values higher than 3, representing a region with low degree of mixing, while the $K_{RCS'}^{unc}$ obtained from 532 nm is composed by values higher and lower than 3. From 1300 m to 3500 m of altitude, the profiles of these two wavelengths are very similar, with values lower than 3 in the region below the *ABLH*, characterizing a well-mixed region, a peak of values higher than 3 in the first meters above the *ABLH* and values between 3 and 4 in the remaining of the profile. The corrections do not cause significant changes in 1064 nm kurtosis profile, as can be seen in Figures A6.29 and A6.30. However the variation in the

kurtosis profile at 532 nm is remarkable, as presented in Figures A6.32 and A6.33. Thus, it becomes very similar to the 1064 nm profile, mainly with the use of first lag correction. The $K_{RCS'}^{unc}$ obtained from 355 nm does not have the same variations observed in the profiles obtained at the reference wavelength. Therefore, it is not possible to identify the occurrence of the phenomenon previously described. The same problem occurs in the $K_{RCS'}^{corr}$, although the application of corrections causes relevant variations in relation to values observed in $K_{RCS'}^{unc}$.

Figure A7 shows the profiles of $\beta_{mol}$, $\beta_{mol+aer}$ and $\beta_{ratio}$ of the wavelengths 1064 nm (Figure A7.1 and 2), 532 nm (Figure A7.3 and 4) and 355 nm (Figure A7.5 and 6). Such profiles were obtained from the data retrieved during the period of analysis presented previously. From the figure A7.1 it is possible to observe the predominance of $\beta_{aer}$ in the wavelength 1064 nm, because of it, the $\beta_{ratio}$ presented in Figure A7.2 achieved large values. In the figure A7.3 it is possible to observe the predominance of $\beta_{aer}$ in the wavelength 532 nm, and a small impact of $\beta_{mol}$. The backscatter profile at 355 nm presented

in figure A7.5 shows that both, $\beta_{aer}$ and $\beta_{mol}$, have the same order of magnitude, however with predominance of $\beta_{aer}$ .





Such profiles justify the differences and similarities observed in the results obtained from each wavelength. Although the backscatter profiles at 532 nm are composed by the molecular and aerosol signatures, the predominance of the last one enables the observation of the phenomenon presented by profiles obtained from the reference wavelength. The small presence of $\beta_{mol}$ also can be an indicator of the low values of noise, although they are higher than the values of reference wavelength.

## 4.2  Case Study II: $19^{th}$ July 2018

In this case study measurements were gathered with the SPU Lidar station from 12:00 to 21:00 UTC. Figure A8 shows the time-height plot of $RCS_{532}$ during this period. In the beginning of measurement it is possible to observe the presence of an ascending *CBL* covered by a *RL*, which has the top situated at around 1300 m of altitude. At approximately 15:30 UTC the *CBL* breaks up the *RL* and becomes fully-developed, thus, its growth speed is reduced and the value of top height maintains practically constant (1600 m) from 17:00 UTC until 21:00 UTC. The black dotted box in Figure A8 represents the chosen period to perform the statistical analysis (18:00 – 19:00 UTC).

In the same way of Case Study I, the hypothesis proposed by Pal et al. (2010) is validated from the profiles presented in Figure A9. The profiles of relative humidity and mixing ratio, presented in the Figure A9-A and A9-b, respectively, do not have large variations in the *CBL* below 1200 m of altitude. In addition, the aerosol optical depth related Ångström Exponent time series did not show considerable changes during the whole measurement period, as can be seen in Figure A9-C. For this measurement period the percentage variation of *AE* was no more than 4% and 3% in the spectral range 340-440 nm and 440-675 nm, respectively. Therefore, there are no considerable changes during the whole measurement period, which is a strong indication that there are no aerosol type change throughout the day and the atmospheric conditions are not propitious for particle hygroscopic growth events.

Figure A10 presents the SNR profile of the raw lidar signal of the three wavelengths (1064 nm (red line), 532 nm (green line), and 355 nm (violet line)) during the analyzed period. In the ABL region, all wavelengths have similar profiles with values higher than 1. However, as ABLH approaches, the values of SNR reduce sharply, mainly of the 355 nm. Consequently, in the FT region all profiles have values lower than 1, as expected.

Figure A11 shows a comparison among the *ACF* obtained from the three wavelengths 1064 nm (left), 532 nm (center) and 355 nm (right), between 18:00 and 19:00 UTC, at two heights 1000 m (red line) and 1700 (green line). In the same way of case Study I, the region above *ABLH* (green line) is more influenced by noise than the region situated below this height (red line). The intensity of $ACF_{532}$ and $ACF_{1064}$ are very similar, although the presence of noise in the first one, which is 40% and 46%, below and above ABLH, respectively, is higher than in the last one, 27% and 30%, below and above ABLH, respectively. The $ACF_{355}$ presents a lower intensity value in comparison with the other two wavelengths, and a strong presence of noise below and above the ABLH, 50% and 67%, respectively.

The three high order moments and $\tau_{RCS}$, both corrected by the first lag correction and obtained between 18:00 and 19:00 UTC, are presented in figure A12. The $\tau_{RCS}^{corr}$ for all wavelengths has values higher than 2s from the bottom of profile until the first meters above the $ABLH_{elastic}$ with maximum of $\sigma_{RCS'}^{2}(z)$. Although the values obtained from 1064 nm and 532 nm are almost twice as large as the values generated from 355 nm.





The positive values of $S_{RCS}^{corr}(z)$ of 1064 nm indicate the presence of aerosol updrafts from the bottom of the profile until around 750 m of altitude. From this height until the *ABLH*, the $S_{RCS}^{corr}(z)$ is characterized by negative values, which represents a region with entrainment of clean *FT* air into the *CBL*. In the same way of case study I, there is an inflection point at *ABLH*, which reproduces the transition from negative to positive values, the last ones indicating the presence of aerosol updraft layers

in the first 200 m above the *ABLH*. Such behavior in the region of *ABLH* also was observed by Pal et al. (2010) and McNicholas and Turner (2014) and it can be considered characteristic of convective regime. The $S_{RCS}^{corr}(z)$ obtained from the wavelengths 1064 and 532 nm presents an identical pattern of behavior, demonstrating the occurrence of the same phenomenon. The $S_{RCS}^{corr}(z)$ obtained from the wavelength 355 nm, in the same way of the previous case study, does not exhibit the behavior observed in the reference wavelength, presenting only positive values in the whole profile. Therefore, it is not possible to

identify variations in the aerosol dynamic using 355 nm.

The $K_{RCS}^{corr}(z)$ obtained from the wavelength 1064 nm presents values higher than 3 from the bottom until around 1300 m of altitude, characterizing a region with low degree of mixing. From 1300 m until the *ABLH* the $K_{RCS}^{corr}(z)$ has values lower than 3, that characterize this region as showing a large degree of mixing and more evidently the presence of turbulence. Such behavior occurs mainly due to of entrainment of cleaner air. A few meters above the *ABLH*, the $K_{RCS}^{corr}(z)$ has a great peak,

which occurs due to rare aerosol plumes penetrating at this region. Such behavior also was observed in case study I, as well as, by Pal et al. (2010) and McNicholas and Turner (2014). Above the *ABLH* the profile has values only higher than 3, however, as $\tau_{RCS}^{corr}(z)$ decreases to values close to zero and low values of *SNR* of the RCS'are characteristic of this region, it is not possible to extract conclusive information from $K_{RCS}^{corr}(z)$. In the same way of the comparison performed with other variables, the $K_{RCS}^{corr}(z)$ obtained from the wavelength 532 nm presents similar behavior to profile obtained from 1064 nm, thus, the same

phenomenon can be observed. On the other hand, the $K_{RCS}^{corr}(z)$ obtained from the wavelength 355 nm does not allow observing the behavior detected in the profile obtained from the reference wavelength, because along the whole profile the $K_{RCS}^{corr}(z)$ at 355 nm presents values higher than 3.

Figure A13 shows the composition signal of $\beta_{aer}$ and $\beta_{mol}$, retrieved during the analyzed period of this case study (18:00 – 19:00 UTC) using the Klett-Fernald-Sasano inversion (Klett, 1983, 1985; Fernald, 1984; Sasano and Nakane, 1984), at each

one of the three wavelengths, as well as the $\beta_{ratio}$ calculated using the backscatter profile of aerosol and molecular component (Bucholtz, 1995). From figure A13-1 it is possible to observe that the backscattered signal at 1064 nm has a predominance of $\beta_{aer}$, with almost null values of $\beta_{mol}$. The composition of the backscattered signal at 532 nm is shown in figure A13-3. Although, the component $\beta_{mol}$ has values higher than that ones observed in wavelength 1064 nm, the component $\beta_{aer}$ is predominant in the backscattered signal composition. The backscattered signal at 355 nm, presented in figure A13-5, unlike

the other wavelengths, is predominantly composed by $\beta_{mol}$ and has a low percentage of $\beta_{aer}$.

From the results obtained in both case studies, it is possible to observe the influence of the wavelength in the proposed methodology. The wavelength 1064 nm, considered as our signal reference, has a negligible influence of component molecular, therefore the backscatter signal retrieved at 1064 nm can be considered approximately equal to the backscatter signal retrieved only by the aerosol contribution, $\beta_{1064} \approx \beta_{aer}$. Before, taking into accounting the approximation demonstrated in equation 5

($RCS_{1064} \approx \beta_{1064}$), we can conclude that the range corrected signal retrieved from a lidar at 1064 nm can be considered, in




an good precision, approximately equal to the backscatter signal retrieved at the same wavelength for aerosol components , $RCS_{1064} \approx \beta_{aer}$. Such relation enables the observation of behavior of aerosol plumes from high order moments. In the case of wavelength 532 nm, $\beta_{532}$ is composed by $\beta_{aer}$ and $\beta_{mol}$ ($\beta_{532} = \beta_{aer_{532}} + \beta_{mol_{532}}$), however, as shown in the Figures A8 and A13, there is a predominance of $\beta_{aer}$ . Although the profiles obtained from the wavelength of 532 nm are slightly nosier

than the profiles generated from the reference wavelength data, the phenomena observed from the 1064 nm data also can be observed in 532 nm data, mainly after the application of first lag correction. Consequently the wavelength at 532 nm can be used in the proposed methodology providing satisfactory results. On the other hand, the backscatter at 355 nm is predominantly composed by $\beta_{mol}$ and has a small percentage of $\beta_{aer}$, as presented in figures A8 and A13.

This fact justifies the low quality observed in the results retrieved using the wavelength of 355 nm. As established in equation

3, the turbulent variable is directly associated with $\beta'_{aer}$, but due to low contribution of this component in the backscatter signal at 355 nm, the supposition established in equation 6 cannot be applied. Consequently, the high-order moments obtained from the proposed methodology are noisier and the value of $\tau_{RCS'}(z)$ is almost half of the value obtained from the reference wavelength, both due to influence of $\beta_{mol}$ that presents the stronger contribution to the total backscatter coefficient at this wavelength. Therefore the behavior observed in the profiles generated from the 1064 nm wavelength data can be detected

partially, or even totally suppressed as the complexity of high-order moments increase. In the both case studies were possible to observe that from the third order moment (skewness) the results obtained from the wavelength 355 nm provide misinformation.

## 5   Conclusions

In this paper we performed a comparative analysis about the use of different wavelengths (355, 532 and 1064 nm) in studies about turbulence. The data were acquired with an elastic lidar, from the SPU Lidar station of LALINET, by measurements

gathered with high frequency (0.5 Hz) along July 2017 to July of 2018. The RCS provided by this system was used to calculate high-order moments (variance, skewness and kurtosis) and the integral time scale, which were applied to characterization of aerosol dynamics. Based on previous studies, the wavelength 1064 nm was adopted as reference due to predominance of $\beta_{aer}$.

Two case studies ($26^{th}$ July 2017 and $19^{th}$ July 2018) were performed in order to verify the proposed methodology, as well as, the applicability of each wavelength. In both cases, the results obtained from 1064 nm wavelength demonstrate as the

high-order moments can support a detailed analysis of the ABL region. In addition, it is remarkable the values of $\tau_{RCS}$ in the region below the *ABLH*, demonstrating the viability of the proposed methodology. The high-order moments obtained from the wavelength 532 nm are slightly more influenced by the noise than the results obtained from the reference wavelength (the value of noise can be observed by the $ACF_{532}$. However, the same phenomenon observed in the profiles generated from the 1064 nm wavelength can be observed in the profiles generated from the wavelength 532 nm, mainly with the application of first lag

correction. On the other hand, the profiles obtained from 355 nm have a strong presence of noise and, thus, from the third order moment (skewness) the phenomenon presented in the profiles obtained from 1064 nm wavelength cannot be observed in 355 nm profiles.





The analysis of the backscatter signal at each wavelength shows that for both case studies $\beta_{aer}$ is a predominant contribution at 532 nm, while $\beta_{mol}$ is predominant at 355 nm. In this way, the high-order statistics become noisier at 355 nm, and it cannot be applied in the proposed methodology. In contrast, the predominance of $\beta_{aer}$ at 532 nm implicates that this wavelength provides results similar to that obtained at 1064 nm, especially after the application of first lag correction. Consequently, the 532 nm wavelength can be used to apply the proposed methodology, providing results similar to that obtained from 1064 nm wavelength.

The results obtained in this paper show the viability of the proposed methodology and its applicability to the 532 nm wavelength, due to the similarity with results derived at 1064 nm and the evidence of a low $\varepsilon$ influence. In addition, the high-order moments obtained from the SPU Lidar station using an elastic lidar data provided us detailed information about some phenomenon in the *ABL*, allowing us a better comprehension about the aerosol dynamics.

*Author contributions.* This paper received the individual contribution according the following statement, conceptualization by G.A. Moreira, J.L. Guerrero-Rascado and L. Alados-Arboledas; methodology by G.A. Moreira, J.L. Guerrero-Rascado and L. Alados-Arboledas and F.J.S. Lopes; Data aquisition by G.A. Moreira, J.J. Silva, A.A. Gomes; software, formal analysis and investigation by G. A. Moreira and F. J. S. Lopes; writing–original draft preparation by G.A. Moreira; writing–review and editing by G.A. Moreira, J.L. Guerrero-Rascado, L. Alados-Arboledas and F.J.S. Lopes; supervision, project administration by L. Alados-Arboledas and E. Landulfo; funding acquisition by E. Landulfo and L. Alados-Arboledas.

*Competing interests.* The authors declare that they have no conflict of interest.

*Acknowledgements.* This work was supported by the Andalusia Regional Government through project P12-RNM-618 2409, by the Spanish Agencia Estatal de Investigación, AEI, through projects CGL2016-81092-R, CGL2017-90884-REDT and CGL2017-83538-C3-1-R. We acknowledge the financial support by the European Union's Horizon 2020 research and innovation program through project ACTRIS-2 (grant agreement No 621654109). The authors thankfully acknowledge the University of Granada that supported this study through the Excellence Units Program and "Plan Propio. Programa 9 Convocatoria 2013". The authors also would like to thank the support from The National Council for Scientific and Technological Development - CNPQ, for the following projects 152156/2018-6, 432515/2018-6 and 150716/2017-6 and São Paulo Research Foundation-FAPESP grant numbers 2015/12793-0.





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



**Table A1.** Variables applied to statistical analysis of turbulence in APBL region (Lenschow et al., 2000). The sum of subindex of autocovariance function $M_{ij}$ represents the order of it.

| | Without Correction | Correction | Error |
|---|---|---|---|
| **INTEGRAL TIME SCALE** $(\tau)$ | $\displaystyle\int_0^\infty RCS'(t)dt$ | $\displaystyle\frac{1}{\overline{RCS'^2}}\int_{t\to 0}^\infty M_{11}(t)dt$ | $\tau\sqrt{\dfrac{4\Delta M_{11}}{M_{11}(\to 0)}}$ |
| **VARIANCE** $(\sigma^2_{RCS})$ | $\displaystyle\frac{1}{T}\sum_{t=1}^T\left(RCS'(t)-\overline{RCS'}\right)^2$ | $M_{11}(\to 0)$ | $RCS'^2\sqrt{\dfrac{4\Delta M_{11}}{M_{11}\to 0}}$ |
| **SKEWNESS** $(S)$ | $\dfrac{\overline{RCS'}^3}{\sigma^3_{RCS'}}$ | $\dfrac{M_{21}(\to 0)}{M_{11}^{3/2}(\to 0)}$ | $\dfrac{\Delta M_{21}}{\Delta M_{11}^{3/2}}$ |
| **Kurtosis** $(K)$ | $\dfrac{\overline{RCS'}^4}{\sigma^4_{RCS'}}$ | $\dfrac{3M_{22}(\to 0)-2M_{31}(\to 0)-3\Delta M_{11}^2}{M_{11}^2(\to 0)}$ | $\dfrac{4\Delta M_{31}-3\Delta M_{22}-\Delta M_{11}^2}{\Delta M_{11}^2}$ |





**Table A2.** Physical meaning of the high-order moments

| | Physical Meaning |
|---|---|
| **INTEGRAL TIME SCALE** ($\tau$) | It is the time over which the turbulent process are highly correlated to itself |
| **VARIANCE** ($\sigma^2_{RCS}$) | It represents the variability of the aerosol concentration during a determined time. |
| **SKEWNESS** ($S$) | It is a measure of the lack of symmetry of a distribution. The values close to zero indicates that the aerosol particles are evenly distributed. Negative values indicates entrainment of clean FT air into the ABL, what causes negative perturbations. On the other hand, the positive values are associated with the center of the aerosol plumes that are penetrating at determined height. |
| **Kurtosis** ($K$) | It is a measure of the flatness of a distribution. Values lower than 3 represents a time series clustered around a mean value, therefore it characterizes a well-mixed ABL region. On the other hand, values higher than 3 indicates the presence of infrequent deviations in the time series, representing a region with low level of mixing. |

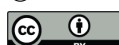



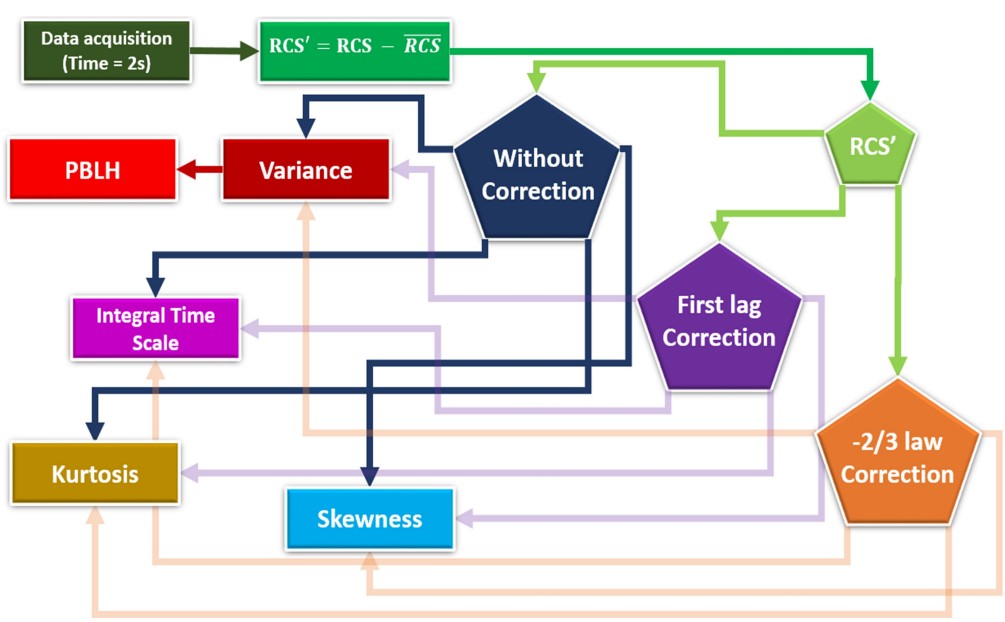

**Figure A1.** Methodological description of data analysis performed for elastic lidar data.



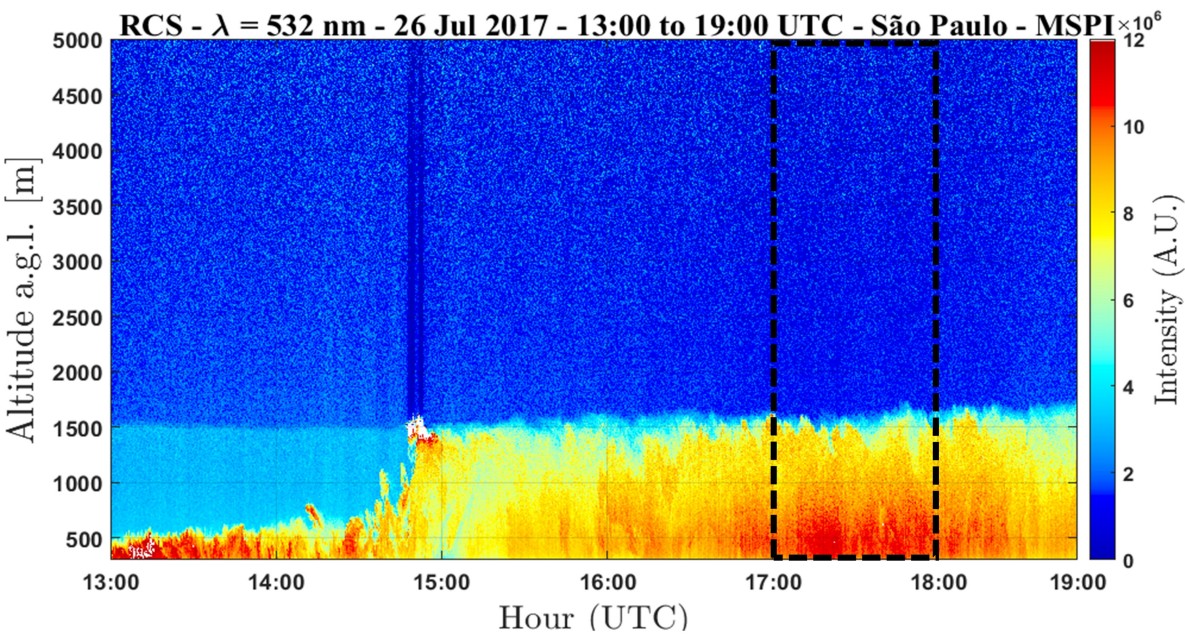

**Figure A2.** Time-Height plot of $RCS_{532}$.





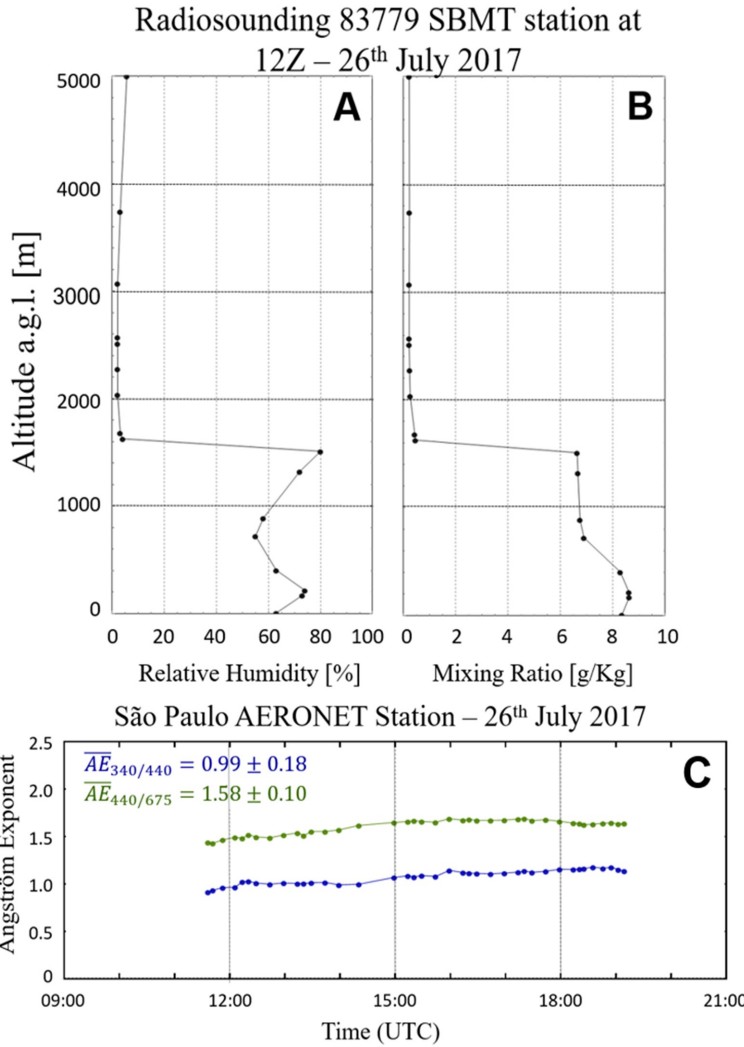

**Figure A3.** (A) Vertical profile of Relative Humidity derived from radiosounding. (B) Mixing Ratio derived from radiosounding. (C) Aerosol optical depth related Ångström Exponent time series from AERONET, for mesaurements retrieved at $26^{th}$ July 2017.





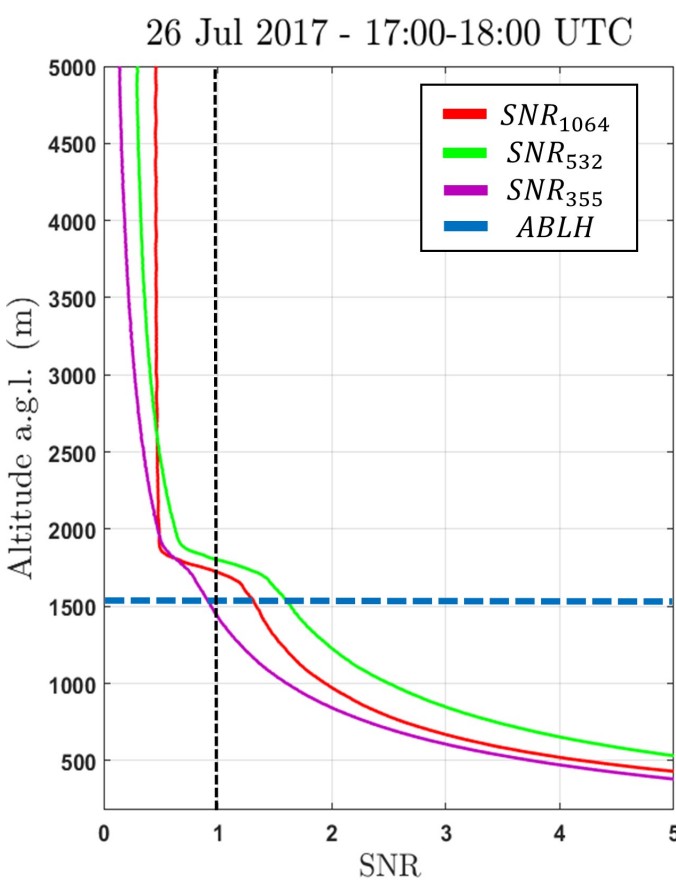

**Figure A4.** Signal-to-Noise-Ratio (SNR) profile of the three wavelengths (1064 nm [red line], 532 nm [green line] and 355 nm [violet line]) obtained at $26^{th}$ Jul 2017 between 17-18 UTC.



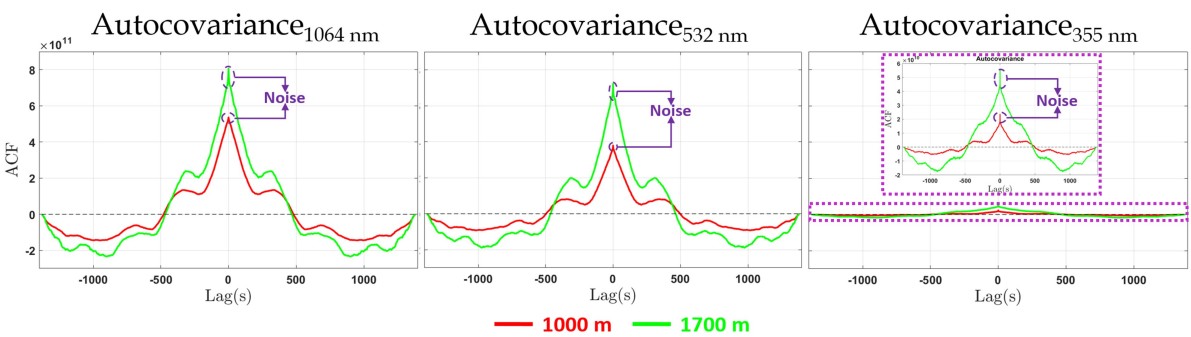

**Figure A5.** Autocovariance function at 1064 nm (left), 532 nm (center) and 355 nm (right) on $26^{th}$ July 2017 from 17:00 to 18:00 UTC. For 355 nm the insert magnifies the signal 10 times.







**Figure A6.** High-Order Moments and $\tau$ without correction and corrected by 2/3 law and first lag correction, for 1064 (red line), 532 (green line) and 355 nm (violet line) on $26^{th}$ July 2017 from 17:00 to 18:00 UTC. The dotted blue horizontal line represents the $ABLH_{elastic}$.




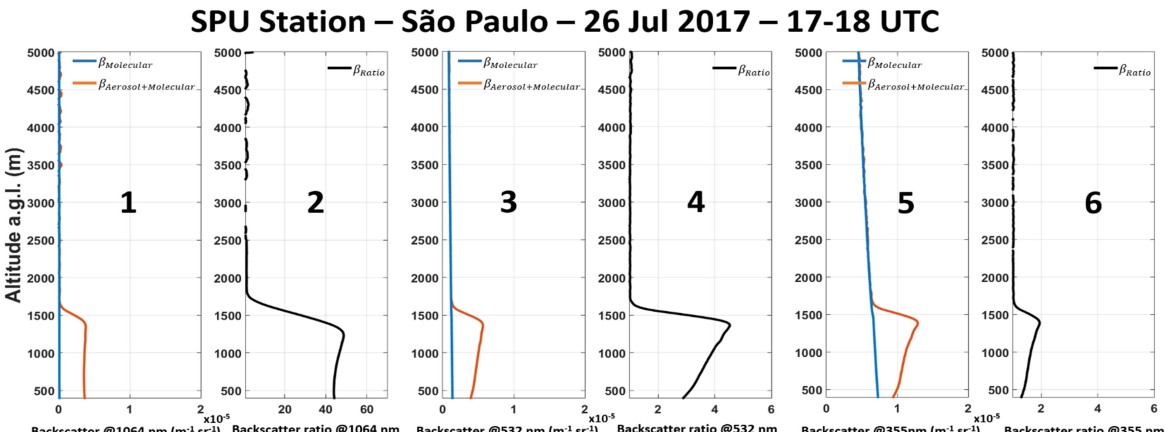

**Figure A7.** Total (aerosol and molecular) backscatter profile and backscatter ratio retrieved using Klett-Fernald-Sasano inversion technique for 1064, 532 and 355 nm, respectively, for data retrieved on $26^{th}$ July 2017 – 17:00-18:00 UTC by the SPU Lidar system.





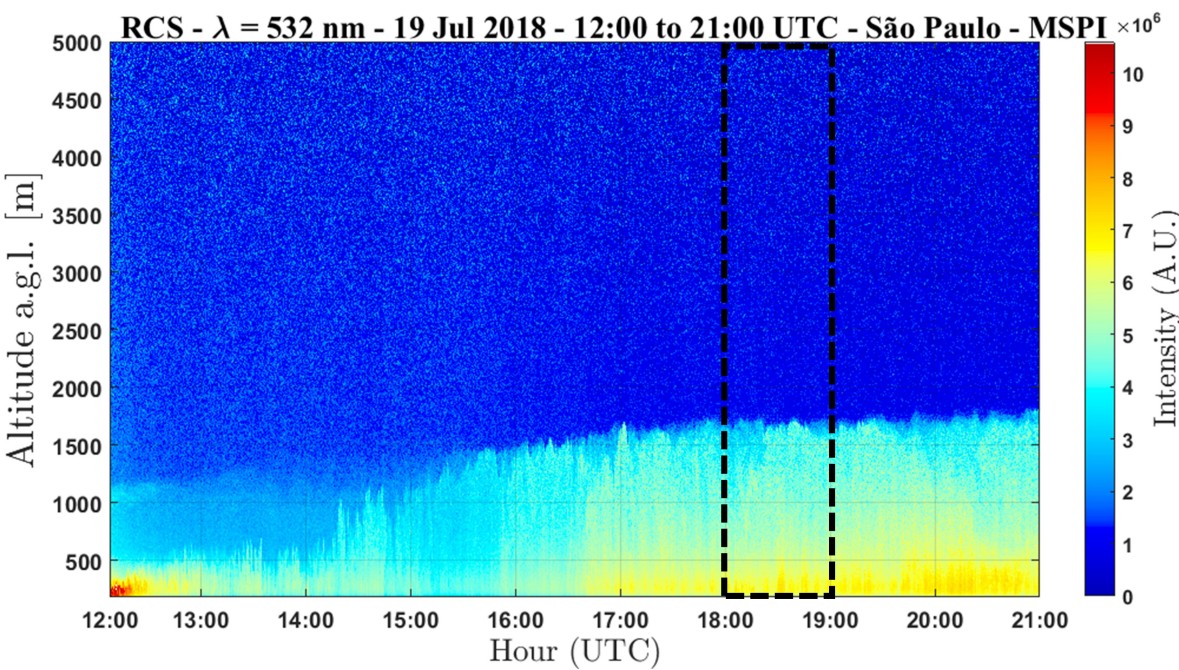

**Figure A8.** Time-Height plot of $RCS_{532}$.



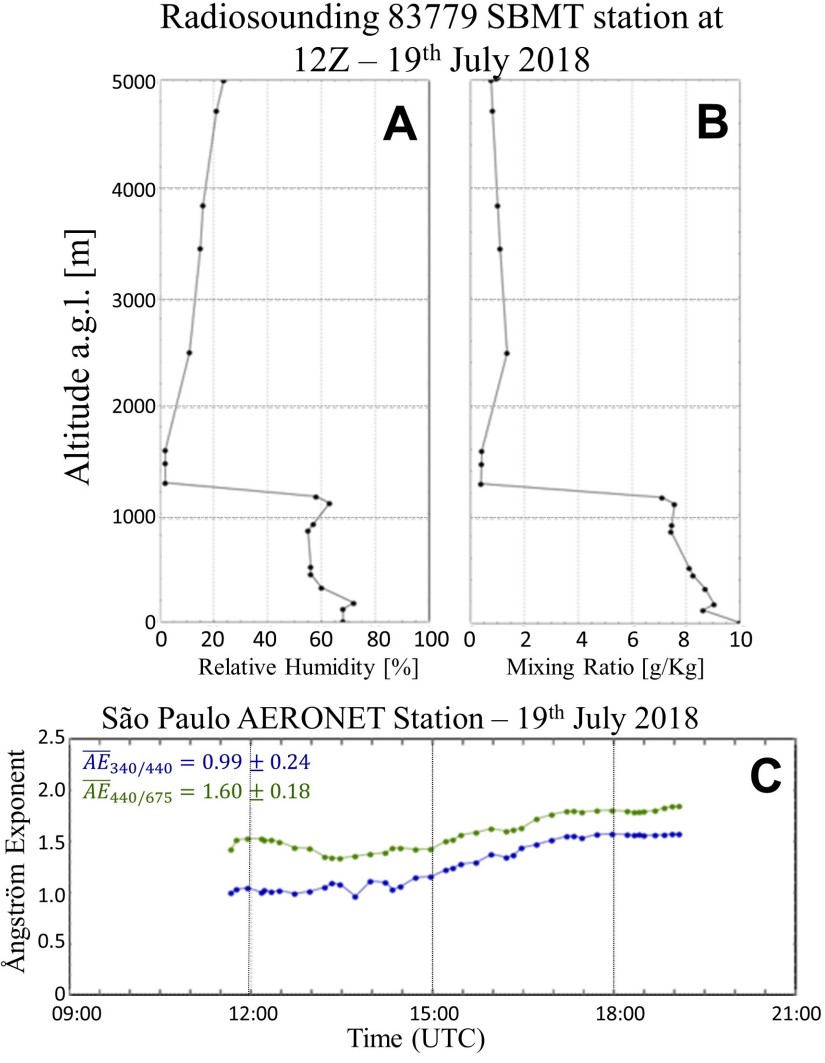

**Figure A9.** (A) Vertical profile of Relative Humidity derived from radiosounding. (B) Mixing Ratio derived from radiosounding. (C) Aerosol optical depth related Ångström Exponent time series from AERONET, for mesaurements retrieved at $19^{th}$ July 2018.





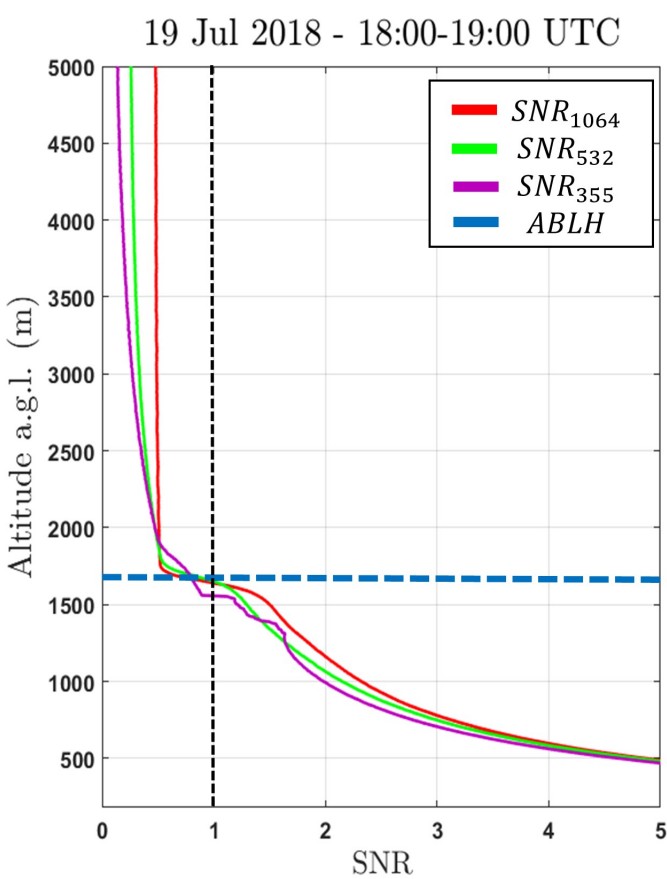

**Figure A10.** Signal-to-Noise-Ratio (SNR) profile of the three wavelengths (1064 nm [red line], 532 nm [green line] and 355 nm [violet line]) obtained at $19^{th}$ Jul 2018 between 18-19 UTC.





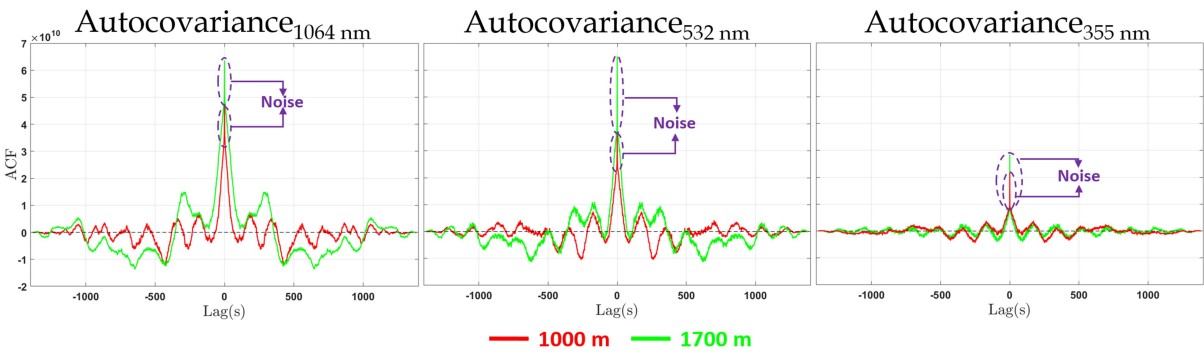

**Figure A11.** Autocovariance function at 1064 nm (left), 532 nm (center) and 355 nm (right) on $19^{th}$ July 2018 from 18:00 to 19:00 UTC.





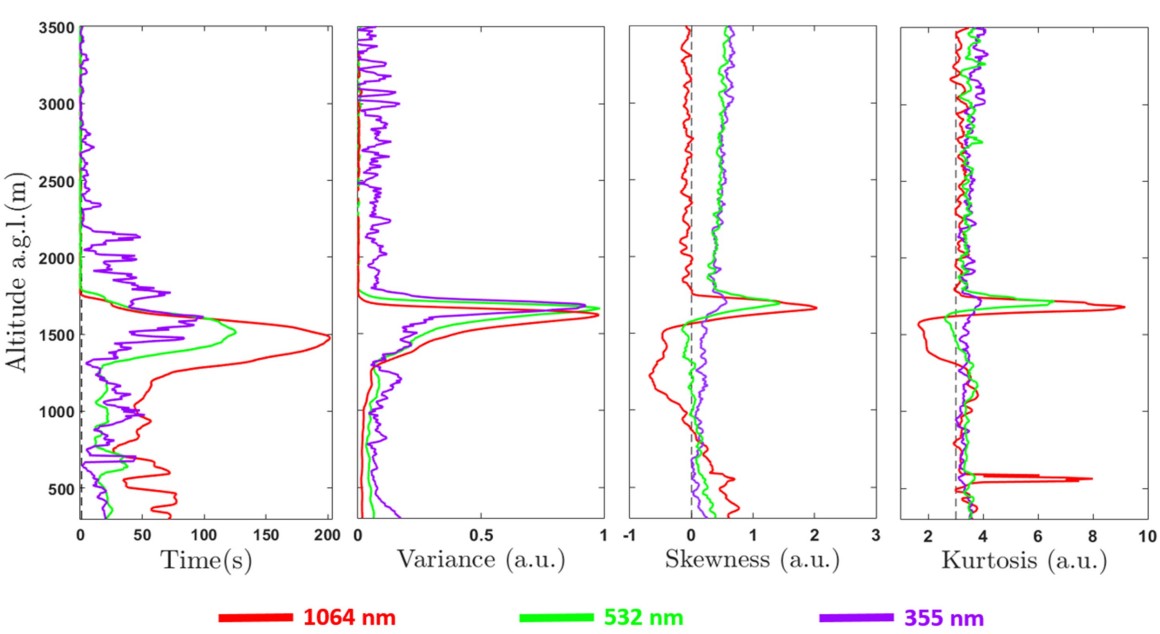

**Figure A12.** High-Order Moments and corrected by first lag correction at 1064 (red line), 532 (green line) and 355 nm (violet line) on $19^{th}$ July 2018 from 18:00 to19:00 UTC.

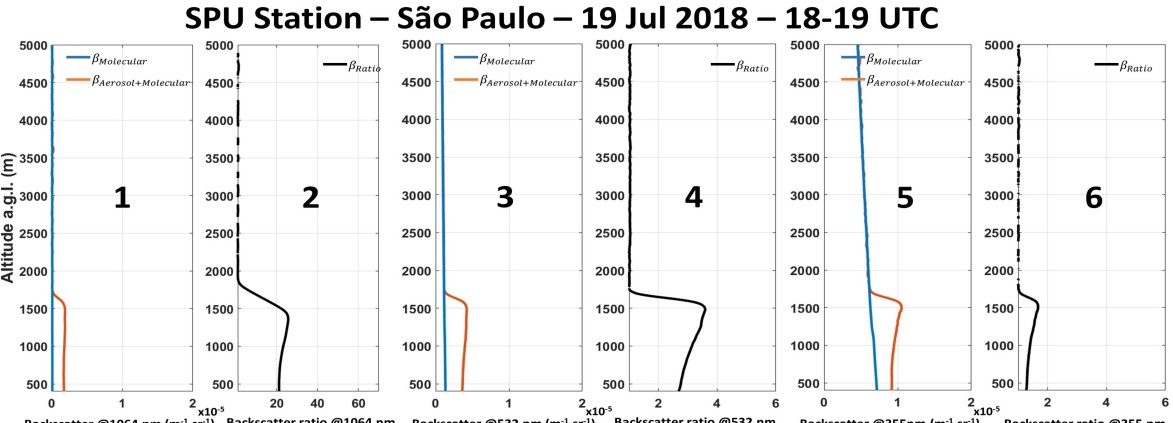

**Figure A13.** Total (aerosol and molecular) backscatter profile and backscatter ratio retrieved using Klett-Fernald-Sasano inversion technique for 1064, 532 and 355 nm, respectively, for data retrieved on $19^{th}$ July 2018 from 18:00 to 19:00 UTC.