# Peer review of "Analyzing the Atmospheric Boundary Layer by high-order moments obtained from multiwavelength lidar data: impact of wavelength choice"

_Atmospheric Measurement Techniques, 2019_

## Referee Comment (RC1) · Anonymous Referee #1 · 19 Apr 2019

General Comments:

The manuscript presents an assessment of three-wavelength lidar measuring PBL turbulence, particularly for the wavelength comparison in term of the high-order moments analysis. Two cases studies are investigated. The physical fundamentals of this study are from the previous work by Pal et al (2010); and by using aerosols as tracers, the PBL turbulence information is illustrated. The current paper needs further information about the methodology and the discussions on the results. There are some typos or grammar errors.

[Figure]

Specific comments:

1. Please give the main specifications of lidar, e.g. laser pulse energy, beam pointing stability, laser pulse repetition rate, detectors and data acquisition. Can you please show the range-corrected signals or images at 1064-nm and 355-nm as Fig.2A?

2. Page-3, Line-12, ". . .from July 2018 to July 2018. . ."? The manuscript only shows two cases studies, not the dataset or measurements from July 2018 to July 2018.

3. Page-4, Eq.(2) and Line 1-3 about the relationship between the aerosol backscatter and number density. Please mention the Mie-theory and aerosol hygroscopic properties with the relative humidity (RH). Under what value of RH, the aerosol hygroscopic properties may be ignored.

4. Page-4, What is the difference between the Eq.(2) and Eq.(3)?

5. Page-4, Eq.(6). Please describe or give the condition(s) or assumption(s) for deriving this equation.

6. Page-6, Line-24, ". . . same type of aerosol is present in the entire atmospheric column . . .". You may assume it for the PBL aerosols, but please note that aerosol type generally depends on both the size distribution and chemical compounds. Thus, it is much different in the near surface, PBL, free troposphere and stratosphere.

7. Page-7, Line 2-4 about the Fig.A3C. The aerosol angstrom exponent can help classify aerosol type in term of aerosol size information. However, it is generally not enough for the different species of aerosols. For instance, both urban aerosols and smoke aerosols are fine-mode particles (i.e. large Angstrom exponents), but they are different types with the different backscatter and extinction properties.

8. Page-7, Line 10-12, the sentence is confused. Why talked about the Figure A2 here? How can you get the first height situated below the top of CBL and the last one at FT from Fig.A2? "As expected tau increases with height for all the wavelengths due to reduction of aerosol load with height. . .."? I can't find it from the figure.
9. Page 8, Line 31-35. Why do you choose the value of 3 as a threshold ("lower than 3 representing a well-mixed region and larger than 3 representing a low degree of mixing")? Figure A5 (28-35) shows the wavelength dependence of the kurtosis profile (KRCS), thus a single threshold seems so arbitrary.

10. Page 10, Line 15-18 about the high-order moments of lidar backscatter signals (skewness and kurtosis). A negative ScorrRCS represents the downdraft while a positive value represents the updraft. Are there any other vertical wind measurements to demonstrate it?

11. Page-10, Line 19-20, "The Scorr RCS(z) obtained from the wavelengths 1064 and 532 nm presents identical pattern of behavior, demonstrating the occurrence of same phenomenon." However, in the Figure A10., they show different and altitude-dependent positive or negative values at 1000-1500-m. For instance, the values at 1064-nm are negative ("downdraft") at 1500-1000m while the values at 532-nm are near zeros. They show different patterns. Why do you call "identical pattern of behavior"?

12. With the low clouds or residual aerosol layers, can the methodology (high-order moments) in this study be applied? Are the high-order moments sensitive to the time window length (e.g. 1-hour long in this paper, 17:00-18:00 UTC for the 1st case, and 18:00-19:00 UTC for the 2nd case)?

Technical corrections or typos:

Page-1, Line-6, "aerosol layers moviments (skewness). . .". moviments or movement?

Page-2, Line-5, "air surface temperature", surface air temperature?

Page-2, Line-8, the meaning of this sentence is confused.

Page-3, Line-14, ". . . located at installed "", some typo.

Page-3, Line-17, "SPU"? full name?

Page-3, Line-18 and 19, please add the unit for the wavelength "387 and 407".

Page-4, Line 16-17, "we can considered. . .". The "considered" should be "consider".

---

## Referee Comment (RC2) · Anonymous Referee #2 · 24 Apr 2019

1- General comments

The paper focuses on atmospheric boundary layer height (ABLH) retrievals and variability analysis from multi-wavelengths lidar measurements. The study highlighted by the authors rely mainly on high-order moments technique developed and descibed in Pal et al, 2010. The current paper emphasizes the importance of wavelength choice used in the technique. Then comparison and discussion on the results obtained applying the technique with lidar measurements from different wavelengths are presented. Two cases are considered in this study to illustrate and to investigate ABLH retrievals

using the described technique. Some paper references dedicated to the related field are missing. The paper presents also some inacurrate wording that need to be revised.

2- Specific comments

- Page 1, line 6-7-8. Why asserting that previous studies have shown that 1064-nm wavelength provides an appropriate description of the turbulence field which is the reason why you consider this wavelength as a reference? Several other papers, prior and since Pal, 2010 have shown related studies for ABLH retrievals that uses different techniques and different wavelengths, including in the UV domain, applied to lidar measurements: Sawyer, et al, 2013, Detection, variations and inter-comparison of the planetary boundary layer depth from radiosonde, lidar and infrared spectrometer http://dx.doi.org/10.1016/j.atmosenv.2013.07.019 Pal et al, journal of geophysical research: atmosphere, vol. 118, 9277–9295, doi:10.1002/jgrd.50710, 2013 Martucci etal, 2007, Comparison between Backscatter Lidar and Radiosonde Measurements of the Diurnal and Nocturnal Stratification in the Lower Troposphere DOI: 10.1175/JTECH2036.1 Wang et al, Atmos. Meas. Tech., 5, 1965–1972, 2012 www.atmos-meas-tech.net/5/1965/2012/ doi:10.5194/amt-5-1965-2012

- Page 5, section 31.1 and Page 22, table A2. Detailed description of high-order moment parameters are given. However, you do not present how ABLH is retrieved from thiese parameters as it is shown in diagram A1 and figures A6 and A10.

-Page 12, line 30. Following discussion about autocorrelated function, you conclude that the profiles obtained at 355nm have a strong presence of noise and thus the skewness phenomenon are not as well retrieved at 355nm compared to those at 1064nm. I assume the authors use the term "profiles"to point out the feature of the autocorrelated function and not the one of the lidar backscatter. Nonetheless, the authors should be more precise.

-page 12, conclusion. The authors conclude that the high-order moments technique is applicable to 532nm elastic lidar measurements and shows results for ABLH retrievals

as well and good as for 1064nm. On the contrary, due to limited validity of the assumption of predominence of aerosol backscatter compared to molecular ones, the retrievals at 355nm are not successfull due to noisier signals. The readers are left a bit curious. It would be usefull for the authors to conclude wheather or not the high-order technique shows limitation for 355nm signal or if the current lidar system used for thes study that could be improved or if the technique should be improved using a better assement of molecular backscatter at 355nm.

- Page 25 & 29, Figue A6 and A10. I do not know why only one ABLH is retrieved since the high-order moments techique is applied for each wavelength independently? I expected to find different retrievals for each wavelegthd and discussion about which one shoud be considered as the truth.

3- technical corrections

- Page 5, line 20, equation (7). the authors shloud define "tf" variable.

- Page 7, line 19. The authors do not define FT. I assume that it means Free Troposphere. You should precise it.

- Page 11, line 34. replace "taking into accounting" by "taking into account"

- Page 20, Figure A1. The diagram indicated PBLH that should be ABLH to be coherent.

---

## Author Comment (AC1) · 2 Jul 2019

We thank the anonymous reviewers for their comments, corrections and suggestions, which have helped to improve the quality of the manuscript. According to the reviewers' reports, the following changes have been performed on the original manuscript and a point-by-point response is included below.

1. Please give the main specifications of lidar, e.g. laser pulse energy, beam pointing stability, laser pulse repetition rate, detectors and data acquisition. Can you please

show the range-corrected signals or images at 1064-nm and 355-nm as Fig.2A?

In order to clarify this point, the text has been changed to provide lidar specifications as follows:

(Page 3, Lines 22-30) "The São Paulo Lidar station (SPU) has a coaxial ground-based multiwavelenght Raman lidar system operated at LEAL. The system operates with a pulsed Nd:YAG laser, emitting radiation at 355, 532 and 1064 nm, a laser repetition rate of 10 Hz and a laser beam pointing to zenith direction. The pulse energy (and stability) of each wavelength are 225 mJ (2mJ) at 355 nm, 400 mJ (4 mJ) at 532 nm, and 850 mJ (6 mJ) at 1064 nm. The MSPI lidar detects three elastic channels at 355, 532 and 1064 nm and three Raman-shifted channels at 387 nm, 408 nm (corresponding to the shifting from 355 nm by N2 and H2O) and 530 nm (corresponding to the Raman shifting from 532 nm by N2). This system is equipped with photomultipliers Hamamatsu R7400. The SPU lidar reaches full overlap at around 300 m a.g.l. (Lopes et al., 2018). This system operates with a temporal and spatial resolutions of 2 s and 7.5 m, respectively."

The following figures has been added as supplementary material:

* Figure C1

* Figure C2

2. Page-3, Line-12, ". . .from July 2018 to July 2018. . ."? The manuscript only shows two cases studies, not the dataset or measurements from July 2018 to July 2018. We thank the Reviewer#1 for this comment. We performed a campaign from July 2017 to July 2018, however only the two best cases are presented in order not to extend the paper much more and to approach in more detail some specific cases. In order to clarify this point the text has been changed as follow:

(Page 3, Lines 17-18)

". . .from July 2017 to July 2018; however, to illustrate the analysis, only two cases are discussed in detail in this article."

3. Page-4, Eq.(2) and Line 1-3 about the relationship between the aerosol backscatter and number density. Please mention the Mie-theory and aerosol hygroscopic properties with the relative humidity (RH). Under what value of RH, the aerosol hygroscopic properties may be ignored. We thank the Reviewer#1 for this comment. In order to clarify this point, the text has been changed as follow:

(Page 4, Lines 7 - 17)

"In the analysis performed with elastic lidar systems, the variable of interest is the aerosol number density (N), from which we obtain its fluctuation (N') by the equation 1. However, elastic lidar systems do not provide directly the value of N. Therefore, considering the validity of Mie-theory (where the aerosol backscatter coefficient is linked to the backscatter efficiency, particle radius (r) and the number of particles with radius r we can write the equation 2, under several assumptions. The premises adopted here are (i) the variation of aerosol size with the relative humidity can be neglected, (ii) the atmospheric volume probed is composed by similar types of aerosol particles and (iii) the fluctuations of the aerosol microphysical properties are smaller than the fluctuations of the total number density in the volume probed by the lidar. More details about these assumptions can be found in (Pal et al., 2010). Feingold (2003) and Titos (2016) demonstrated the relation between relative humidity and hygroscopic growth, so that, such effects can start at 80% RH. The two cases presented in this work were gathered in winter, the driest season of São Paulo. In particular, RH was below 80% in both days (see section 4). Such value is lower than the RH threshold to hygroscopic effects indicated by the two papers above mentioned."

Feingold, Graham: Aerosol hygroscopic properties as measured by lidar and comparison with in situ measurements. Journal of Geophysical Research, Vol. 108, 4327, doi:10.1029/2002JD002842, 2003.

Titos, G., Cazorla, A., Zieger, P., Andrews, E., Lyamani, H., Granados-Muñoz, M. J., Olmo, F. J., Alados-Arboledas, L.: Effect of hygroscopic growth

on the aerosol light-scattering coefficient: A review of measurements, techniques and error sources. Atmospheric Environment, Vol. 141, 494-507, https://doi.org/10.1016/j.atmosenv.2016.07.021, 2016.

4. Page-4, What is the difference between the Eq.(2) and Eq.(3)?

We thank the Reviewer#1 for this question. Eq. (2) presents the relationship between particle backscatter coefficient ($\beta$_aer) the aerosol number density (N). On other hand, Eq. (3) presents the relationship between the fluctuations of these same variables. The fluctuations are obtained from the Reynold's decomposition (Eq. 1).

5. Page-4, Eq.(6). Please describe or give the condition(s) or assumption(s) for deriving this equation. We thank the Reviewer 1 for this question. The Eq. 6 is obtained from Reynold's decomposition (Eq. 1). In order to clarify this point, the text has been changed as follow:

(Page 5, Line 11)

"...particles. Then, applying Reynold's decomposition (Eq. 1) over Eq. 5, the following equation is derived:..."

6. Page-6, Line-24, ". . . same type of aerosol is present in the entire atmospheric column . . .". You may assume it for the PBL aerosols, but please note that aerosol type generally depends on both the size distribution and chemical compounds. Thus, it is much different in the near surface, PBL, free troposphere and stratosphere. We thank the Reviewer 1 for this comment. In order to adjust this point, the text has been changed as follow:

(Page 7, Line 18)

"...the same type of aerosol is present in the entire atmospheric column in the ABL region..."

7. Page-7, Line 2-4 about the Fig.A3C. The aerosol angstrom exponent can help clas-

sify aerosol type in term of aerosol size information. However, it is generally not enough for the different species of aerosols. For instance, both urban aerosols and smoke aerosols are fine-mode particles (i.e. large Angstrom exponents), but they are different types with the different backscatter and extinction properties.

The Angstrom Exponent can give an indication of the aerosol type, however, it is not enough to classify different types of aerosol. Recently, a new method to aerosol classification was presented by Papagiannopoulos et. al., 2018) based on the intensive optical parameters retrieved from the European Aerosol Research Lidar Network (EARLINET). The predictive accuracy of this automatic classification method varies between 59'% to 90'% (maximum) applied to 8 to 4 aerosol classes. In order to apply this aerosol classification method the author used typical lidar configuration for the EARLINET lidars, a multi-wavelength Raman lidars combining a set of elastic and inelastic channels, the so-called $3\beta+2\alpha$ configuration, in addition of polarization channels. Using or apply the aerosol classification method is not the aim of this work. Furthermore, it would not be possible considering that our lidar configuration, a multi-wavelength Raman lidar ($3\beta+2\alpha$) with no depolarization, was set up in 2018. However, we used the 530 nm rotational Raman channel (Veselovskii et al, 2015) to the lidar ratio profile for the case of 29 of July 2018. As can be seen in the following figure, we could retrieve the lidar ratio (LR) profile up to 1500 m, as the ground level, which correspond the altitude of the atmospheric boundary layer. The LR oscillate around the mean lidar ratio of $53 \pm 7$ sr, which is a strong indication that there is no changes in the aerosol optical properties during the turbulence analysis period. Unfortunately, for the case of 26 of July 2017, it was not possible to retrieve the lidar ratio profile since we did not have the rotational Raman lidar configuration. However, since both cases are very similar when comparing the Relativity humidity, the mixing ratio, and the temporal distribution of the Angstrom exponent, we can assume that there are no considerable changes during the whole measurement period for 26 of July 2017.

* Figure C3

Papagiannopoulos, N., Mona, L., Amodeo, A., D'Amico, G., Gumà Claramunt, P., Pappalardo, G., Alados-Arboledas, L., Guerrero-Rascado, J. L., Amiridis, V., Kokkalis, P., Apituley, A., Baars, H., Schwarz, A., Wandinger, U., Binietoglou, I., Nicolae, D., Bortoli, D., Comerón, A., Rodríguez-Gómez, A., Sicard, M., Papayannis, A., and Wiegner, M.: An automatic observation-based aerosol typing method for EARLINET, Atmos. Chem. Phys., 18, 15879-15901, https://doi.org/10.5194/acp-18-15879-2018, 2018.

Veselovskii, I., Whiteman, D. N., Korenskiy, M., Suvorina, A., and Pérez-Ramírez, D.: Use of rotational Raman measurements in multiwavelength aerosol lidar for evaluation of particle backscattering and extinction, Atmos. Meas. Tech., 8, 4111-4122, https://doi.org/10.5194/amt-8-4111-2015, 2015.

8. Page-7, Line 10-12, the sentence is confused. Why talked about the Figure A2 here? How can you get the first height situated below the top of CBL and the last one at FT from Fig.A2? "As expected tau increases with height for all the wavelengths due to reduction of aerosol load with height. . ..."? I can't find it from the figure. C2 We thank the Reviewer#1 for this comment. In order to clarify these points, the text has been changed as follows:

(Page 7, Line 23)

"Thus, from the comparison of the figures A2 and A5 it is possible to observe that the altitude chosen at 1000 m (red line) is situated below the top of CBL, while the altitude chosen at 1700 m (light green line) is in the FT. As expected, the $\varepsilon$, which is represented by the peak on the lag 0 of the autocovariance function (fig.5), increases with height. . ."

9. Page 8, Line 31-35. Why do you choose the value of 3 as a threshold ("lower than 3 representing a well-mixed region and larger than 3 representing a low degree of mixing")? Figure A5 (28-35) shows the wavelength dependence of the kurtosis profile (KRCS), thus a single threshold seems so arbitrary. We thank the reviewer#1 for this comment. The kurtosis equation in the table A1 represents the kurtosis of a Normal

Distribution (ND), which is equal 3 (Bulmer, 1965). Therefore, due to we observe the variations of the high-order moments referred to ND, the value 3 is adopted as threshold for our kurtosis analyses. In order to clarify these points, the text has been changed as follows:

(Page 10, Line 3-5)

"The kurtosis equation presented in the table A1 represents the kurtosis of a Normal Distribution, which is equal 3 (Bulmer, 1965), consequently such value is applied as threshold in the analyses performed in this paper.

Bulmer, M. G., Principles of Statistics, 1965.

10. Page 10, Line 15-18 about the high-order moments of lidar backscatter signals (skewness and kurtosis). A negative $S\_corr^{RCS}$ represents the downdraft while a positive value represents the updraft. Are there any other vertical wind measurements to demonstrate it?

Unfortunately, we do not have measurements of the vertical wind speed collocated to the SPU lidar. However, the conceptual definition of skewness (a measure of the asymmetry of the probability distribution) enable us to confirm that a negative $S\_corr^{RCS}$ represents the downdraft while a positive value represents the updraft, and this fact was validated during the SLOPE-I campaign, which was performed with elastic and Doppler lidar in Granada-Spain during the summer of 2016 (Bedoya-Velásques et al., 2018, Moreira et. al, 2019).

Bedoya-Velásquez, A. E., Navas-Guzmán, F., Granados-Muñoz, M. J., Titos, G., Román, R., Casquero-Vera, J. A., Ortiz-Amezcua, P., Benavent-Oltra, J. A., de Arruda Moreira, G., Montilla-Rosero, E., Hoyos, C. D., Artiñano, B., Coz, E., Olmo-Reyes, F. J., Alados-Arboledas, L. and Guerrero-Rascado, J. L.: Hygroscopic growth study in the framework of EARLINET during the SLOPE I campaign: synergy of remote sensing and in situ instrumentation. Atmospheric Chemistry and Physics, 18, 10, 7001-7017,

http://doi.org/10.5194/acp-18-7001-2018, 2018.

11. Page-10, Line 19-20, "The Scorr RCS(z) obtained from the wavelengths 1064 and 532 nm presents identical pattern of behavior, demonstrating the occurrence of same phenomenon." However, in the Figure A10., they show different and altitude-dependent positive or negative values at 1000-1500-m. For instance, the values at 1064-nm are negative ("downdraft") at 1500-1000m while the values at 532-nm are near zeros. They show different patterns. Why do you call "identical pattern of behavior"?

We thank the reviewer#1 for asking this question. The main goal of the turbulence analysis by aerosol lidars is to provide a characterization about the phenomena (like as direction of vertical movements and/or level of mixing), and not its absolute values. Therefore, in this kind of analysis the turbulence cannot be estimated but only characterized. Although the skewness of 1064 nm presents absolute values higher than that observed from 532 nm, in both cases the vertical pattern of skewness is the same in the most of the profile and, therefore, the same phenomena can be observed in both profiles. This is the reason why we affirm an identical pattern of behavior.

12. With the low clouds or residual aerosol layers, can the methodology (high-order moments) in this study be applied?

We thank the Reviewer 1 for this question. Yes, such methodology can be applied in the presence of clouds or aerosol residual layers. The presence of low clouds provides results where a predominance of cloud-driven turbulence can be observed. In the cases where the residual layer is present, it is possible to observe its interaction with the CBL and FT. Examples of these situations are shown in Moreira et. al (2019).

In order to clarify these points, the text has been changed as follows:

(Page 7, Line 5 - 6) "Examples of the application of such methodology in varied meteorological scenarios (presence of clouds and aerosol sublayers) are presented in Moreira et al. (2019)."

Are the high-order moments sensitive to the time window length (e.g. 1-hour long in this paper, 17:00-18:00 UTC for the 1st case, and 18:00-19:00 UTC for the 2nd case)?

Yes, as the window length is reduced the integral time scale is affected, what can reduce the region of atmospheric column where the lidar system can solve the high-order moments. Based on our elastic lidar resolution and earlier papers (Paul et al., 2010 and McNicholas et al., 2014) we decide to use the time window of 1 hour. Considering the lidar system used in this paper, small time windows do not enable us to estimate the high order moments in the whole PBL region. In order to clarify these points, the text has been changed as follows:

(Page 6, Line 23)

"one-hour (the influence of time-window is demonstrated in Moreira et al. (2019))"

Technical corrections or typos:

Page-1, Line-6, "aerosol layers moviments (skewness). . .". moviments or movement? ". . .aerosol layers movement. . ."

Page-2, Line-5, "air surface temperature", surface air temperature? ". . .surface air temperature. . ."

Page-2, Line-8, the meaning of this sentence is confused. The sentence has been rewritten as presented below: "Slightly before sunset, the decrease of the incoming solar irradiance at the surface results in a radiative cooling of the Earth's surface."

Page-3, Line-14, ". . . located at installed ", some typo. The sentence has been rewritten as presented below: "This lidar facility is installed at the Nuclear and Energy. . ."

Page-3, Line-17, "SPU"? full name? The sentence has been rewritten as presented below: "The São Paulo Lidar station (SPU). . ."

Page-3, Line-18 and 19, please add the unit for the wavelength "387 and 407". The sentence has been rewritten as presented below: "... three Raman-shifted channels at

387 nm, 408 nm (corresponding to the shifting of 355nm by N2 and H2O). . ."

[Figure]

[Figure]

Figure C1. Time Height plot of

**Fig. 1.**

[Figure]

Figure C2. Time Height plot of

**Fig. 2.**

[Figure]

Figure C3. Backscatter, extinction and Lidar ratio profile retrieved using Rotational Raman lidar analysis for 19 of July 2018.

**Fig. 3.**

---

## Author Comment (AC2) · 2 Jul 2019

We thank the anonymous reviewers for their comments, corrections and suggestions, which have helped to improve the quality of the manuscript. According to the reviewers' reports, the following changes have been performed on the original manuscript and a point-by-point response is included below.

Specific comments

Page 1, line 6-7-8. Why asserting that previous studies have shown that 1064-

nm wavelength provides an appropriate description of the turbulence field which is the reason why you consider this wavelength as a reference? Several other papers, prior and since Pal, 2010 have shown related studies for ABLH retrievals that uses different techniques and different wavelengths, including in the UV domain, applied to lidar measurements: Sawyer, et al, 2013, Detection, variations and inter-comparison of the planetary boundary layer depth from radiosonde, lidar and infrared spectrometer http://dx.doi.org/10.1016/j.atmosenv.2013.07.019 Pal et al, journal of geophysical research: atmosphere, vol. 118, 9277–9295, doi:10.1002/jgrd.50710, 2013 Martucci etal, 2007, Comparison between Backscatter Lidar and Radiosonde Measurements of the Diurnal and Nocturnal Stratification in the Lower Troposphere DOI: 10.1175/JTECH2036.1 Wang et al, Atmos. Meas. Tech., 5, 1965–1972, 2012 www.atmos-meas-tech.net/5/1965/2012/ doi:10.5194/amt-5-1965-2012

We thank the Reviewer 2 for this question. In this paper, we describe the turbulence field of aerosols, so that the wavelength 1064 nm is the most convenient due to the practically null contribution of molecular signal to this channel, what enable us to perform the simplification shown in section 3 (equation 2), in agreement with results shown in Pal et al., 2010. Although others wavelengths, like as UV and 532 nm, can be applied efficiently in the ABLH detection (papers recommended by the referee), they have a higher contribution of molecular signal in comparison with 1064nm, what can increase the noise in the high-order moments as shown in section 3.2. In order to clarify this point, the text has been changed as follow:

(Page 1, Line 6-8)

"Previous studies have shown that 1064-nm wavelength, due to the predominance of particle signature in the total backscattered atmospheric signal and practically null presence of molecular signal (which can represent noise in high-order moments), provides an appropriate description of the turbulence field and thus..."

(Page 2, Line 25-26)

[Figure]

"...remote sensing systems (mainly lidars) become an important tool in ABLH detection (Martucci et al., 2007; Pal et al., 2010; Wang et al., 2012), as well as, in turbulence studies..."

Page 5, section 31.1 and Page 22, table A2. Detailed description of high-order moment parameters are given. However, you do not present how ABLH is retrieved from these parameters as it is shown in diagram A1 and figures A6 and A10.

We thank the Reviewer 2 for this question. At the top of the CBL there is entrainment of clear air masses coming from FT, causing fluctuations in the aerosol's concentration in this region and consequently in the RCS profile. Therefore, the height with maximum variance in the RCS profile can be used as indicator of the ABLH. This methodology is named Variance Method and has the limited applicability only for convective cases. In order to clarify this point, the text has been changed as follow:

(Page 7, Line 3-4)

"The ABLH is estimated from the Variance Method, which establish, in convective conditions, the top of CBL (ABLH) as the maximum of the variance of the RCS [$\sigma\_RCS^2 (z)$](Baars et al., 2008)."

Page 12, line 30. Following discussion about autocorrelated function, you conclude that the profiles obtained at 355nm have a strong presence of noise and thus the skewness phenomenon are not as well retrieved at 355nm compared to those at 1064nm. I assume the authors use the term "profiles" to point out the feature of the autocorrelated function and not the one of the lidar backscatter. Nonetheless, the authors should be more precise.

We thank the reviewer#2 for this comment. We apologize due to the generalization of term "profile". Such attitude becomes some parts of the text very confusing. In order to clarify this point, the text has been changed as follow:

(Page 1, Line 12)

"...the noise associated to the high-order profiles..."

(Page 10, Line 29)

"... phenomenon presented by high-order moments profiles obtained from..."

(Page 12, Line 32-33)

"Although the high-order moments profiles obtained from the wavelength 532 nm are noisier than that one generated from..."

(Page 13, Line 8)

"Therefore the behavior observed in the high-order moments profiles generated from..."

(Page 13, Line 23-24)

"However, the same phenomena observed in the high-order moments profiles generated from the 1064 nm wavelength can be observed in that one generated from the wavelength 532 nm..."

(Page 13, Line 25-27)

"On the other hand, the high-order moments obtained from 355 nm have a strong presence of noise and, thus, from the third order moment (skewness) the phenomenon presented in the high-order moments obtained from 1064 nm wavelength cannot be observed in 355 nm high-order moments profiles. ."

Page 12, conclusion. The authors conclude that the high-order moments technique is applicable to 532nm elastic lidar measurements and shows results for ABLH retrievals C2 as well and good as for 1064nm. On the contrary, due to limited validity of the assumption of predominance of aerosol backscatter compared to molecular ones, the retrievals at 355nm are not successful due to noisier signals. The readers are left a bit curious. It would be useful for the authors to conclude weather or not the high-order

technique shows limitation for 355nm signal or if the current lidar system used for this study that could be improved or if the technique should be improved using a better assessment of molecular backscatter at 355nm.

We thank the Reviewer 2 for this comment. The proposed methodology is based on the utilization of particle signal, which is strongly present in 1064 and 532 nm wavelength. The wavelength 355 nm has a predominance of molecular signal, this is the reason of its inapplicability in the proposed methodology. However, a better assessment of the molecular backscatter at 355 can reduce the influence of the noise caused by molecular signal and improve the results obtained from the data generated from this channel. In order to clarify this point, the text has been changed as follow:

(Page 14, Line 2-5)

"On the other hand, the wavelength 355 nm does not provide satisfactory results in such methodology due to predominance of molecular signal in its composition. However, a better assessment of the molecular backscatter at 355 can reduce the influence of the noise caused by molecular signal and improve the results obtained from the data generated from this channel."

Page 25 & 29, Figure A6 and A10. I do not know why only one ABLH is retrieved since the high-order moments technique is applied for each wavelength independently? I expected to find different retrievals for each wavelength and discussion about which one should be considered as the truth.

We thank the Reviewer 2 for this comment. The two selected days have a well-defined ABL with high similarity among the RCS profiles generated from the three wavelengths, as can be observed in figures C1 and C2 of supplementary material, so that, the ABLH obtained from Variance Method to each wavelength is practically the same with difference lower than 10% as can be observed in figures A6 and A12. In order to clarify this point, the text has been changed as follow:

[Figure]

(Page 8, Line 32-34)

"Although the $\sigma\_355^2$ is nosier than another ones, there is a low difference among the ABLH estimated from the three different wavelengths (lower than 10%)."

(Page 11, Line 27-29)

"In the same way of Case I, although there are some differences among the maximum of the $\sigma\_RCS^2$ (z), they do not influence significantly the ABLH estimation, so that, the difference among the ABLH obtained from each wavelength is lower than 10%."

Technical corrections

Page 5, line 20, equation (7). the authors should define "tf" variable. Done Page 6, Line 5: "where tf means final time."

Page 7, line 19. The authors do not define FT. I assume that it means Free Troposphere. You should precise it. Done Page 8, Line 4: "... of values lower than 1 in the Free Troposphere (FT), what was..."

Page 11, line 34. replace "taking into accounting" by "taking into account" Done

Page 20, Figure A1. The diagram indicated PBLH that should be ABLH to be coherent. In order to clarify this point the figure A1 has been redone as shown below:
* * *
[Figure]

Data acquisition (Time = 2s) → $RCS' = RCS - \overline{RCS}$ → RCS'

Without Correction

Variance → ABLH

Integral Time Scale

Kurtosis

Skewness

First lag Correction

-2/3 law Correction

**Fig. 1.**

---

## Author Response (AR1)

**Analyzing the Atmospheric Boundary Layer by high-order moments obtained from multiwavelenght lidar data: impact of wavelength choice *by* Gregori de Arruda Moreira et al.**

**Author's response.**

We thank the anonymous reviewers for their comments, corrections and suggestions, which have helped to improve the quality of the manuscript. According to the reviewers' reports, the following changes have been performed on the original manuscript and a point-by-point response is included below.

**Reviewer** 1**

**General Comments:**

The manuscript presents an assessment of three-wavelength lidar measuring PBL turbulence, particularly for the wavelength comparison in term of the high-order moments analysis. Two cases studies are investigated. The physical fundamentals of this study are from the previous work by Pal et al (2010); and by using aerosols as tracers, the PBL turbulence information is illustrated. The current paper needs further information about the methodology and the discussions on the results. There are some typos or grammar errors.

**Specific comments:**

1. Please give the main specifications of lidar, e.g. laser pulse energy, beam pointing stability, laser pulse repetition rate, detectors and data acquisition. Can you please show the range-corrected signals or images at 1064-nm and 355-nm as Fig.2A?

In order to clarify this point, the text has been changed to provide lidar specifications as follows:

**(Page 3, Lines 22-30)**

"The São Paulo Lidar station (SPU) has a coaxial ground-based multiwavelenght Raman lidar system operated at LEAL. The system operates with a pulsed Nd:YAG laser, emitting radiation at 355, 532 and 1064 nm, a laser repetition rate of 10 Hz and a laser beam pointing to zenith direction. The pulse energy (and stability) of each wavelength are 225 mJ (2mJ) at 355 nm, 400 mJ (4 mJ) at 532 nm, and 850 mJ (6 mJ) at 1064 nm. The MSPI lidar detects three elastic channels at 355, 532 and 1064 nm and three Raman-shifted channels at 387 nm, 408 nm (corresponding to the shifting from 355 nm by N2 and H2O) and 530 nm (corresponding to the Raman shifting from 532 nm by N2). This system is equipped with photomultipliers Hamamatsu R7400. The SPU lidar reaches full overlap at around 300 m a.g.l. (Lopes et al., 2018). This system operates with a temporal and spatial resolutions of 2 s and 7.5 m, respectively."